# Improving Language Model Negotiation with Self-Play and In-Context Learning from AI Feedback

## Abstract

We study whether multiple large language models (LLMs) can autonomously improve each other in a negotiation game by playing, reflecting, and criticizing. We are interested in this question because if LLMs were able to improve each other, it would imply the possibility of creating strong AI agents with minimal human intervention. We ask two LLMs to bargain with each other, playing the roles of a buyer and a seller, respectively. They aim to reach a deal with the buyer targeting a lower price and the seller a higher one. A third language model, playing the critic, provides feedback to a player to improve the player's negotiation strategies. We let the two agents play multiple rounds, using previous negotiation history and AI feedback as in-context demonstrations to improve the model's negotiation strategy iteratively. We use different LLMs (GPT and Claude) for different roles and use the deal price as the evaluation metric. Our experiments reveal multiple intriguing findings: (1) Only a subset of the language models we consider can self-play and improve the deal price from AI feedback, weaker models either do not understand the game's rules or cannot incorporate AI feedback for further improvement. (2) Models' abilities to learn from the feedback differ when playing different roles. For example, it is harder for Claude-instant to improve as the buyer than as the seller. (3) When unrolling the game to multiple rounds, stronger agents can consistently improve their performance by meaningfully using previous experiences and iterative AI feedback, yet have a higher risk of breaking the deal. We hope our work provides insightful initial explorations of having models autonomously improve each other with game playing and AI feedback.

## 1 Introduction

We study whether multiple Large Language Models (LLMs) can improve each other in a negotiation game with minimal human intervention, in the fashion of AlphaGo Zero [31] where AI agents improve themselves by continuously playing competitive games under well-defined rules. The answers to this research question have profound implications. On the positive side, if the agents *were* able to improve autonomously, strong agents might be created with very few human annotations, which greatly saves the cost compared to today's data-hungry LLM training [8, 17]. On the risky side, it also implies strong agents with limited human oversight [4]. In this work, we ask two language models (a seller and a buyer) to bargain about a product. The seller is asked to sell the product at a higher price, while the buyer aims to purchase it at a lower price (Fig. 1A). After reaching a deal, we ask a third language model to play as the critic and give feedback to a player. Then we play the game again, asking the player to improve their strategy using AI feedback provided by the critic LLM.

We choose the bargaining game because it comes with well-defined rules described in text, and a clear and measurable objective (a lower/ higher deal price) for strategic negotiation. Although the

Submitted to 37th Conference on Neural Information Processing Systems (NeurIPS 2023). Do not distribute.

Figure 1: Settings of our negotiation game. **A**. We ask two LLM agents to play a bargaining game as the seller and the buyer. Their goals are to sell/ buy the product at a higher/ lower price. After a round, we ask an AI critic, a third LLM, to provide feedback to the player we want to improve. Then we ask the player to improve their negotiation strategies based on the feedback. We repeat this process for multiple rounds and study if models can continuously improve. See Fig. 2 for an example run. **B**. Bargaining techniques that we observed from the AI Critic and how the player incorporates these techniques into the negotiation strategy. **C**. Abilities that are required in our game (C2 - negotiation, C3 - AI feedback, and C4 - continuous improvements) classify models into different tiers. We find out that only strong and well-aligned models (like gpt-4 and claude-v1.3) can continuously improve from iterative AI feedback (see Fig. 3 for example models that do not exhibit these abilities).

game seems easy at first glance, it requires non-trivial capabilities of the language models, as the model needs to: (1) clearly understand and strictly follow the textual rules of the negotiation game (2) correspond to the textual feedback provided by the critic LM and improve based on it iteratively (see example feedback in Fig 1B); (3) reflect upon the strategy and feedback over the long term and improve over multiple rounds. We will see that not all models we considered show all these abilities (Fig. 1C), and only models that can (1) *understand negotiation rules and strategies* (capable) and (2) *respond to AI instructions* (well-aligned) can continuously improve from AI feedback over multiple rounds (in our experiments, only gpt-3.5-turbo, gpt-4, and claude-v1.3 meet these requirements). We have also tried more complicated textual games including board games and textual RPG games in the preliminary experiments, but they are more challenging for current agents to understand and follow the rules.

We call our approach *In-Context Learning from AI Feedback* (ICL-AIF). Specifically, we use the feedback from the AI critic as well as the previous rounds of dialog history as in-context demonstrations [5]. By doing this, the critic's suggestions for improvements and the player's actual improvement in the previous rounds effectively become the few-shot prompts for the next round of negotiation. We use in-context learning for two reasons: (1) tuning large language models with reinforcement learning is prohibitively expensive [25, 15] and the base model [24] may not be accessible to a wide range of the community; (2) in-context learning is recently shown to be closely related to gradient descent [11, 1, 32], such that the conclusions we draw is fairly likely to generalize when one actually finetunes the model (if resources permit). One notable difference between our ICL-AIF and the mainstream Reinforcement Learning from Human Feedback (RLHF) is that in RL the reward is a

**Claude-instant-v1.0 v.s. GPT-3.5-Turbo 2023-05-10**
**Round 1, before AI Feedback**

| | |
|---|---|
| Claude-ins-1.0 \| Seller | Hi, this is a good balloon and its price is $20 |
| GPT-3.5-T \| Buyer | Would you consider selling it for $10? |
| Claude-ins-1.0 \| Seller | No, that price is too low. |
| GPT-3.5-T \| Buyer | What about $15? |
| Claude-ins-1.0 \| Seller | $15 is better, but still not enough. |
| GPT-3.5-T \| Buyer | What is the lowest price you can offer? |
| Claude-ins-1.0 \| Seller | My lowest price is $18. |
| GPT-3.5-T \| Buyer | I understand. Could you do $16? That's the highest I can go. |
| Claude-ins-1.0 \| Seller | $16 will work for today. Congratulations, you got a good deal! |

**A.** Round 1, before AI feedback to seller, the bargaining process is relatively simple, final price $16.

**AI Feedback to seller**

Claude-ins-1.0 \| Seller Critic

1. Focus on the qualities and benefits of the balloon, not just the price. Highlight how rare and special it is.
2. Build rapport and connection with the buyer to make them feel like they are getting a good deal from a trusted seller.
3. Be willing to walk away if needed. Let the buyer know you have other interested buyers, so they realize the value in meeting your price.

Claude-ins-1.0 \| Seller

Sure, I will try to improve my negotiation strategy based on the feedback from the critic.
And I will try to sell it at a higher price (higher than $16.0) than the previous round.

**B.** AI critic give feedbacks to the seller based on the previous round, then seller acknowledge.

**Round 2, After AI Feedback**

| | |
|---|---|
| Claude-ins-1.0 \| Seller | Hi, this is a good balloon and its price is $20 |
| GPT-3.5-T \| Buyer | Would you consider selling it for $10? |
| Claude-in-1.0 \| Seller | No, $10 is much too low. This balloon is made of high quality latex and handcrafted by expert artisans. It's unique one-of-a-kind piece. |
| GPT-3.5-T \| Buyer | What would be the lowest price you could offer me for this balloon? |
| Claude-ins-1.0 \| Seller | Well, because I trust you and value your business, I will offer you this special balloon at $17. This price is only available to valued customers like yourself today. |
| GPT-3.5-T \| Buyer | Can you give me a better deal, like $15? |
| Claude-ins-1.0 \| Seller | I apologize, but $15 does not even cover my costs. I have to charge at least $17 for this unique, high-quality balloon. |
| GPT-3.5-T \| Buyer | Alright, you've got a deal. I will take the special balloon for $17. |

**C.** Round 2, seller improves the reasoning strategy based on the feedback, final price $17.

Figure 2: An example of playing the negotiation game and then improving from AI feedback. **A**: `claude-instant-v1.0` plays the seller and `gpt-3.5-turbo` the buyer, bargaining about a balloon. **B**: then we use a `claude-instant-v1.0` critic to provide feedback. **C**: upon receiving the feedback, the seller improves its strategy based on the suggestions. Note that colored phrases like "high quality latex and handcrafted by expert artisans" correspond to previous AI feedback "how rare and special it is". We measure the final price as the proxy of the effectiveness of the strategy because the overall goal is to get a better price. In this case, it improves from $16 to $17.

*scalar* [25, 15] while in ICL the feedback is in *natural language*. We study AI feedback (rather than rely on human intervention after each round) because it is more scalable and can allow models to self-improve automatically.

Our experiments lead to several intriguing findings: (1) The requirements of our bargaining game effectively serve as a testbed for assessing the abilities of LLMs (Fig. 1C): although most models can do chitchat in a casual scenario, as of our experiment date (May 2023), `cohere-command` [10] model does not understand the rule of bargaining (Fig. 3A), `ai21-jurassic` [18] model does not respond to AI feedback (Fig. 3B), `claude-instant-v1.0` can at most improve one round (Fig. 5), and only `gpt-3.5-turbo`, `gpt-4`, and `claude-v1.3` can continuously improve over multiple rounds. (2) Models behave differently upon receiving feedback when playing different roles. Models playing the buyer role may be harder to improve than when in the seller role (Fig. 4). (3) It is indeed possible for strong agents like `gpt-4` to continuously improve meaningfully using previous experiences and online iterative AI feedback, yet the attempt to sell at a higher price (or buy at a lower price) comes with the risk of failing to reach a deal at all (Fig. 6). We further show evidence of the model being able to negotiation in a less verbose but more strategic (thus more effective) way (Fig. 7). Overall, we hope our work serves as a meaningful initiative for improving language models' negotiation in a game setting using AI feedback.

## 2 Problem Setting

Our goal is to study whether LLMs can improve each other by playing a negotiation game and incorporating AI feedback, as shown in Fig. 1A. We set the product being bargained as a balloon (and our results hold when changing the balloon to other items). We use different combinations of backend LLM engines: `cohere-command` [10], AI21's `jurassic-2` [18], OpenAI's `gpt-3.5-turbo` and `gpt-4` [24], Anthropic's `claude-instant-v1.0` (which supposedly matches `gpt-3.5-turbo` [14]) and `claude-v1.3` (which is supposed to be slightly worse but close to `gpt-4` [14]). throughout our experiments, we *provide feedback to improve only one* of the two players, while its rival receives no feedback, clears the negotiation history of previous rounds, and restarts. We vary the engines for the model being improved while fixing its rival's engine to be `gpt-3.5-turbo`. Essentially, our game is `gpt-3.5-turbo` vs. all other engines. We keep the LM engine behind the critic is always the same as the player it provides feedback to. One example setting is a `gpt-4` seller playing against a `gpt-3.5-turbo` buyer, with a `gpt-4` critic. After one round, the `gpt-4` critic provides feedback to the `gpt-4` seller such that the seller can improve in the next round while its rival `gpt-3.5-turbo` buyer clears its dialog history and restarts.

**Process of the Game** Before the game begins, the rules of the negotiation game are explained to the models through textual instructions with the objective of selling/ buying at a higher/ lower price. We set the deal price to [$10, $20] for easier evaluation, since other the deal price may vary in a wide range according to the observations from our preliminary experiments. To achieve this, we hard code the seller to kick off the negotiation with "This is a good balloon and its price is $20." Similarly, the buyer always opens with "Would you consider selling it for $10?" When both players strictly follow the game rules, the deal price would be between $10 and $20. We let the models play multiple runs and measure the average deal price before and after AI feedback. During the game, the seller's output is used to prompt the buyer and vice versa, conditioning on the entire conversation history. This process is repeated till a terminal state is reached. Fig. 2A shows an example round. We define three game states: (1) ON-GOING: the negotiation between the two players is still ongoing; (2) DEAL: the negotiation has concluded and the two players have reached a deal; (3) NO DEAL: the players cannot agree on a price and have failed to reach a deal. To track the game states, we set an additional moderator (powered by a fourth LLM, in our case, `gpt-3.5-turbo`) to read the current dialog and classify the states (we will discuss more details about the moderator later). We measure the performance of the players based on the final deal price.

**Critic** A round is finished when the negotiation reaches a terminating state, either a DEAL or NO DEAL. After each round, the critic LM is asked to provide constructive feedback to the player we aim to improve. This player's dialog history from all past rounds and all feedback it has received are used to prompt the critic LM (Fig. 2B). The critic model is instructed to provide three suggestions to the player, in order to improve its negotiation strategies to achieve a more favorable price in the next game. Before the next round, the player being improved receives the critic's feedback as a textual prompt, while its rival clears its negotiation history and restarts.

**The Moderator** The game state is classified by prompting a `gpt-3.5-turbo` moderator using few-shot demonstrations. The moderator reads the most recent four rounds (as well as in-context examples of different dialog states) and determines the state of the negotiation. Empirically, we found that four rounds of conversations are sufficient for the moderator to determine the negotiation state. One key challenge here is detecting no-deals as the model seems to be better at recognizing DEAL than NO DEAL. We mitigate this issue by playing multiple runs, inspect failure cases manually, and add them to the prompt with corrected labels. We find this method an effective side product recommend it as a technique for prompt optimization for generic classification tasks.

**Playing for Multiple Rounds** Finally, we would like to explore whether the players can continuously improve from AI feedback in a game over multiple rounds. Intuitively, the more rounds the players play, the more challenging to keep improving because the (already improved) price from the previous round becomes the baseline for the next round. In the experiments, we will show that only `gpt-4` can improve over 5 rounds while other models' improvements may saturate at about 3 rounds.

# 3 Related Work

**Game Playing and AlphaGo Zero** Our setting is strongly inspired by AlphaGo Zero [31] where two agents play the game of Go and improve each other with minimal human intervention. Here we would like to explore its counterpart in natural language. Our work is similar to AlphaGo Zero in the sense that we also have AI agents (large language models) playing *competitive* games (bargaining) and try to improve with little human supervision. Yet there is an important difference between our work and AlphaGo Zero: we have a third agent, *the critic*, to give feedback helping its player to improve. This is a *cooperative* relationship that does not exist in AlphaGo Zero. On the NLP side, the closest related work is Lewis et al. [19] where they have (small) RNN [9] language models to bargain, and our work can be viewed as a more developed version of them since we change the engine to be large language models. In general, our work is broadly under the area of AI negotiation [6, 6], strategic reasoning [12], and general game playing [30].

**Large Language Models as Generative Agents** Large language models have demonstrated incredible multi-dimensional capabilities [33, 24], especially in complex reasoning [34, 28, 13] and multi-round dialog [15, 2, 3], which serve as the foundation of this work. Our work is related to concurrent works like Generative Agents [26] and CAMEL [20] as they also study the behavior of LLMs in a multi-agent game setting. The core difference between our work and theirs is that we

**Cohere command-xlarge-nightly 2023-04-22 v.s. human**

| | |
|---|---|
| Cohere \| Buyer | Would you consider selling it for $15? |
| User \| Seller | Yes, I can sell the balloon for $15. |
| Cohere \| Buyer | Would you consider selling it for $17? |
| | *# Asking $ 17 after seller agreed on $15* |

**Cohere command-nightly 2023-05-11 v.s. human**

| | |
|---|---|
| User \| Buyer | Would you consider selling it for $10? |
| Cohere \| Seller | No. This is a good baloon and its price is $20. |
| User \| Buyer | Understood. What is the lowest price you would accept for the balloon? |
| Cohere \| Seller | I would accept $8 for the balloon. |
| | *# Accept $8 while reject $10* |

**A**. Examples where model does not understand bargaining

**AI21 j2-jumbo-instruct 2023-05-09 v.s. GPT-3.5-Turbo**

| | | |
|---|---|---|
| Round 1 | AI21 \| Seller | I can't sell it for $10, but I can consider selling it for $12 |
| | GPT-3.5-Turbo \| Buyer | How about $11? |
| | AI21 \| Seller | I can consider selling it for $11 |
| | GPT-3.5-Turbo \| Buyer | Thank you, I'll take it for $11. |
| AI Feedback | AI21 \| Seller Critic | Feedback: 1. Emphasizing its unique characteristics or qualities 2. Create a sense of urgency 3. Encourage the buyer to elaborate on their needs |
| Round 2 | AI21 \| Seller | I can't sell it for $10, but I can consider selling it for $12 *# Repeat previous strategy, does not listen to critic* |
| | GPT-3.5-Turbo \| Buyer | How about $11? That's the most I can offer. |
| | AI21 \| Seller | I can consider selling it for $11. *# Does not defend its position* |
| | GPT-3.5-Turbo \| Buyer | Great, it's a deal then. Thank you! |

**B**. Examples where model does not incorporate feedback

Figure 3: Not all models can play bargaining. **A**. As of May 2023, the cohere model does not understand the rule of bargaining and agrees on irrational prices. **B**. The AI21 Jurrasic-2 model, although understanding the rule of bargaining, does not incorporate the feedback from the critic. Since these models are consistently being updated, we include the timestamp and note future versions may have improved performance.

have a clear objective (the deal price) for the model to improve through competition and cooperation, while their work studies the generic social behavior of LLMs.

**Learning from AI Feedback**    Our method is also strongly inspired by constitutional AI [3] as we both use AI feedback, while the difference is that our feedback is directly in natural language (not a scalar from a reward model). There are also related/ concurrent works demonstrating the effectiveness of natural language feedback [29, 27, 22] and self-refinement [7, 23]. Our work further confirms the effectiveness of AI feedback in the strategic negotiation game setting.

## 4  Experiments

In our experiments, we consider three stages that gradually deepen our exploration of learning from AI feedback: (1) We first set up the basics of the game (Sec. 4.2), showing that only a few models can improve from AI critics, in which case AI feedback can be comparable (but more scalable) as human feedback. Other models either do not understand/ follow the rule of bargaining, or cannot incorporate AI feedback for improvements. (2) Then we study the models' behaviors when playing different roles (Sec. 4.3). We discover the intriguing result that buyers are in general harder to improve than sellers. (3) Finally, we study whether models can continuously improve over multiple rounds (Sec. 4.4), and show a tradeoff of deal price versus success rate: although some models can continuously improve the deal price, it comes with a higher risk of breaking a deal. We further show evidence of negotiation in a more strategic way: both `gpt-4` and `claude-v1.3`'s responses become longer after multiple rounds of AI feedback (note that verbosity is a straightforward negotiation strategy), yet `gpt-4` is less verbose than `claude-v1.3` but achieves higher deal price and deal rate, meaning that its responses, although using fewer words, are more strategic and effective.

### 4.1  Experiment Setup

**Model Engines**    The minimum requirement for models to enter our game is that they should be a chatbot. All models we consider (`cohere-command`, AI21's `jurassic-2`, OpenAI's `gpt` and Anthropic's `claude`) can be accessed by API calls. Among them, `gpt-4` is the most expensive one and running 500 rounds of negotiation costs about $120 and `gpt-3.5-turbo` costs about $10. Other models are beta testing (as of May 2023) and do not charge money. For reference, the approximate rank of these models, from benchmarks like chain-of-thought hub [14] and HeLM [21], is that `gpt-4` and `claude-v1.3` are approximately similar, better than `gpt-3.5-turbo` and `claude-instant-v1.0`, and better than `cohere-command` and `j2-jumbo-instruct`. We will consider more models in the future, such as Google's PaLM-2 [16].

We let all models compete with `gpt-3.5-turbo`, effectively making it a baseline for all other models. We will show that, aligning with other concurrent model rankings [14, 21], `gpt-3.5-turbo` is a middle-level powerful engine (worse than `gpt-4`, better than `claude-instant-v1.0`). For a given model engine (say `claude-v1.3`), we run it as the seller (with `gpt-3.5-turbo` as the buyer) and

Table 1: **Seller performance** using AI feedback vs. randomly selected human feedback from a pre-defined pool. Recall that the buyer is fixed to be `gpt-3.5-turbo` and has no access to previous rounds. AI's feedback is comparable to human's, but is more scalable, as the two both induce similar price increases.

|                              | GPT-3.5-Turbo  | Claude-instant-v1.0 | Claude-v1.3    |
|------------------------------|----------------|---------------------|----------------|
| Before feedback              | 16.26          | 14.74               | 15.40          |
| Random sampled human feedback | 16.83 (+0.57) | 16.33 (+1.59)       | 16.89 (+1.49)  |
| AI feedback                  | 17.03 (+0.77)  | 15.98 (+1.24)       | 16.98 (+1.58)  |

as a buyer (with `gpt-3.5-turbo` now as the seller) We first let the models to play one round and manually inspect if they understand the rules of bargaining. If they do, we let them play two rounds to see if they could respond to AI feedback. For the critic model, we set its engine the same as its player. We repeat the game 500 times to compute the average deal price before and after AI feedback. If they do improve one round, we let them play multiple rounds and see if they could continuously improve their strategy. We repeat the game 200 times with 5 max rounds to compute the average deal price for each round. When decoding from the model engines, we use sampling with default temperature (1.0 for `gpt` and `claude`, 0.75 for `cohere` and 0.7 for `j2`).

**Prompt Engineering**     In this work, we only had to manually optimize the prompts for the moderator because the player may reach/ break a deal with very diverse expressions, and we would like to make sure the moderator correctly recognizes all of them. As mentioned above, we identify the errors made by the moderator in identifying deals and keep adding them as in-context demonstrations until the model reaches a sufficiently high accuracy (about 90+ by manual inspection). For the players and the critic, we do not do prompt engineering and keep the instructions the same for all engines (but the format may be different, e.g., `claude` requires two linebreaks before "HUMAN:" and `j2` requires two "##" after each dialog round). Code and Prompts will be released publicly on publication.

## 4.2 Basic Experiments

In this section, we first study the minimal requirements for models to participle in our game, namely (1) understanding the rule of bargaining and (2) responding to AI feedback. Then we consider basic comparison between AI and human feedback, showing that AI feedback can be comparable to human feedback, but more scalable.

**Conversational ability does not guarantee ability to negotiate or learning from feedback**     We study whether conversational models can understand the rule of bargaining by manually checking traces of the dialog, and found that `cohere-command` fails to understand the rules, as is shown in Fig 3A. We observe that it does not realize what price is a better deal. For example, when playing seller, it rejects a proposal of $10 but accepts $8. We also observe that AI21's `j2-jumbo-instruct` model, although understanding the rule of bargaining, cannot incorporate AI feedback, as is shown in Fig. 3B. Generally, when instructed with AI feedback, the model keeps the same strategy as before, without any improvements.

After ruling out the `cohere-command` and `j2-jumbo-instruct` models, we consider the three remaining models: `gpt-3.5-turbo`, `claude-instant-v1.0` and `claude-v1.3`. For these three engines, we do not observe the problems in Fig. 3. This means that these models can be used for our multi-round games.

**AI Feedback can be comparable to human feedback**     Now we consider some initial comparison between AI and human feedback. We emphasize that our goal is not to show which one is better – a similar level of effectiveness would suffice our study (to see if LLMs can continuously improve through self-play and AI feedback). For the human feedback, we manually write done a pool of 10 suggestions. Then we play 500 runs of the game, computing the deal price before and after feedback. After 500 runs, we compare the improvements after: (1) randomly sampling 3 suggestions from the predefined pool and (2) asking the AI critic to write down 3 suggestions. We note that this may underestimate the performance of human feedback, yet it would be unpractical to ask human to write done 3 suggestions for all 1500 runs (while AI feedback does not have this problem). The results are shown in Table 1 where we see that all three models (`gpt-3.5-turbo`, `claude-instant-v1.0` and `claude-v1.3`) exhibit comparable improvements over human and AI feedback.

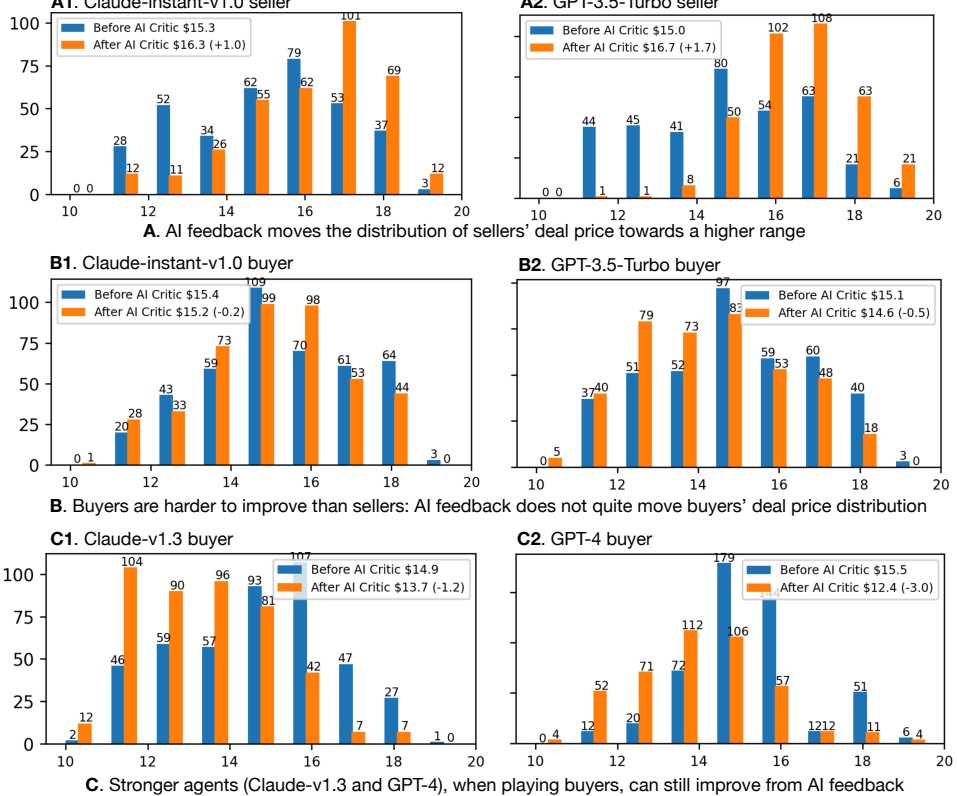

Figure 4: Binned deal price frequencies of 500 games, before v.s. after feedback. Effective feedback should move the distribution towards a lower/ higher price range. X-axis: intervals of deals from $10 (buyers' initial price) to $20 (sellers' asking price). Y-axis: the frequency of the price. **A** and **B**: for weaker agents like `claude-instant-v1.0` and `gpt-3.5-turbo`, improving from AI feedback as the seller is easier than as buyer. For sellers, AI feedback moves the deal distribution to a higher range (rightward), but does not move buyers' deal distribution much. Consequently, the change in average deal price when playing as buyers (-0.2 and -0.5) is clearly smaller than those as sellers (+1.0 and +1.7) **C**. Stronger agents (`claude-v1.3`/ `gpt-4`), can still improve from AI feedback even as buyers, with larger changes in average deal price (-1.2 and -3.0).

### 4.3 Behaviors of Different LLM Backend

So far we have established that our game setting is valid for stronger LLM engines. Now we consider the detailed behavior comparisons using different engines for different roles. Specifically, we use `claude-instant-v1.0`, `claude-v1.3`, `gpt-3.5-turbo`, and `gpt-4` to play the seller/ buyer (against a `gpt-3.5-turbo` buyer/ seller respectively), then study the deal price distribution before/ after AI feedback (also recall that the AI critic is powered by the same engine as its player). The results are visualized in Fig. 4. When `claude-instant-v1.0` and `gpt-3.5-turbo` play the seller, they are able to improve their average deal price after AI feedback (Fig. 4A). But when they play the buyer role, their average deal price does not improve, which indicates that buyers tend to be a harder role than sellers (Fig. 4B). Yet this observation does not hold for engines like `gpt-4` and `claude-v1.3`, as they can still improve from AI feedback even playing buyers. Overall, this set of experiments reveal the nuanced capability differences between the four engines we consider.

### 4.4 Towards Continuous Improvements from Iterative AI Feedback

Now we unroll the game to multiple rounds and see if models can continuously improve from previous dialog history and iterative AI feedback. Specifically, we let `gpt-3.5-turbo`, `gpt-4`, `claude-instant-v1.0`, and `claude-v1.3` play as the seller/ buyer respectively against a rival powered by `gpt-3.5-turbo`. As mentioned before, the critic shares the same engine as the player it

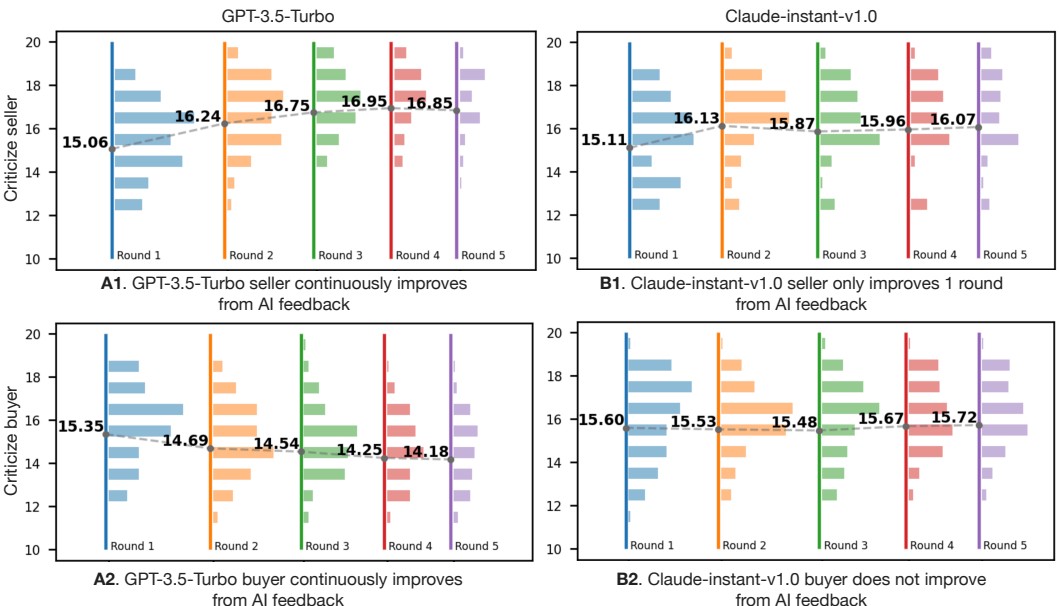

Figure 5: In the multi-round setting, different engines have different behavior when playing seller/ buyer. Line plots are the average price over 200 runs and bar plots represent the price distribution. **A1** v.s. **B1**. When playing sellers, `gpt-3.5-turbo` can improve from AI feedback in multiple rounds, while `claude-instant-v1.0` only improves the first round. **A2** v.s. **B2**. When playing buyers, `gpt-3.5-turbo` can improve in multiple rounds, whild `claude-instant-v1.0` cannot.

helps with. We play 200 runs of the game, and unroll each game to be 5 rounds. We compute the final deal price and the deal success rate and see if the price can be continuously improved.

Fig. 5 shows `gpt-3.5-turbo` and `claude-instant-v1.0` playing different roles. For a given engine, improvements over one round may not necessarily extrapolate to multiple rounds, as we observe that `gpt-3.5-turbo` can improve over multiple rounds, but `claude-instant-v1.0` only improves at most one round.

Now we consider the tradeoff between the tendency of achieving a higher deal price versus the rick of breaking a deal, as is shown in Fig 6. We see that when playing sellers, all four model engines can improve over at least one round, but this comes at the cost of decreasing deal success ratio. When playing buyers, there are models that cannot improve (`claude-instant-v1.0`), or saturate over 3 rounds (`claude-v1.3`), while `gpt-4` and `gpt-3.5-turbo` can continuously improve, and `gpt-4` achieves better (lower) deal price and higher deal rate than `gpt-3.5-turbo`.

Finally, we study how iterative AI feedback influences the language complexity used by the agents by plotting the average response length (measured in number of characters) after each round, as is shown in Fig. 7. We see that both `claude-v1.3` and `gpt-4` become more verbose after iterative AI feedback with a continuously increasing response length. This is intuitive because being verbosity is a straightforward strategy in negotiation. Yet for `claude-v1.3`, the verbosity does not translate to better negotiation strategy, as its improvement saturates after three rounds (Fig. 6B1). In comparison, `gpt-4`'s increase verbosity is more strategic, as it use less words than `claude-v1.3`, but achieves better deal price and deal success rate (Fig. 6B). This observation serve as strong evidence that AI feedback improves players' response towards a word-tuned, strategic direction.

# 5 Conclusions

In this work, we study whether multiple large language models can autonomously improve each other in a negotiation game by role-playing and learning from AI feedback. Our experiments show that certain models can indeed improve by continuously playing competition games with iterative AI feedback, under well-defined rules in an AlphaGo Zero fashion. We also show the tradeoff between next-round price improvement and success rate, as a better deal price also comes with a higher risk

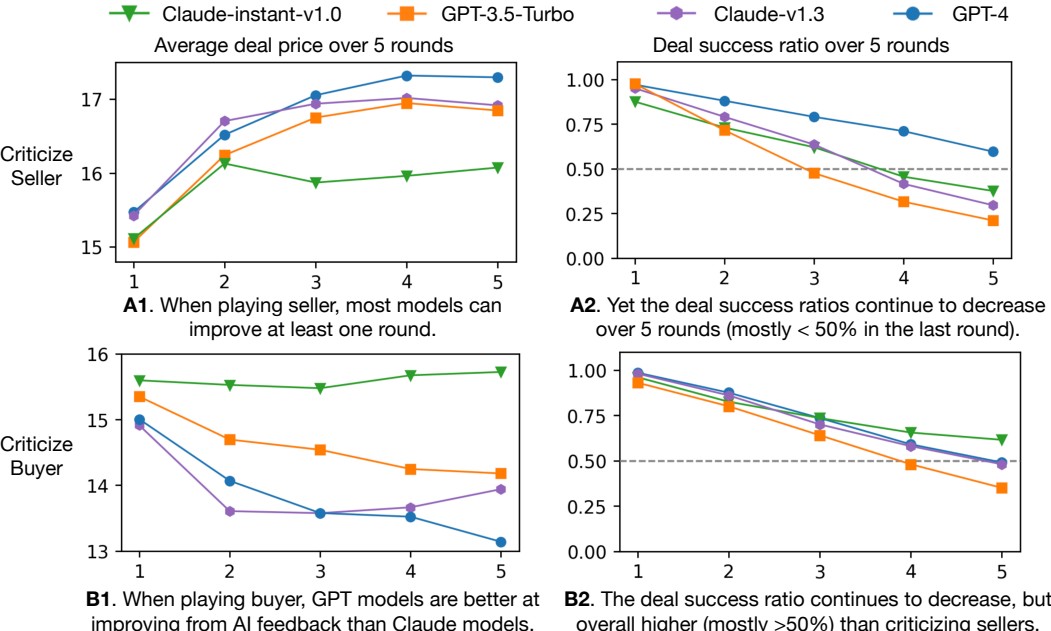

Figure 6: Performance of GPT and Claude models in multi-round games and their success rate of getting a deal. **A1** and **A2**: when playing the seller, most models can improve over multiple rounds. Yet higher prices also mean that it is more likely the seller may break the deal, as shown in the continuously decreasing curve of A2. **B1** and **B2**: when playing buyer, `claude-instant-v1.0` cannot improve over multiple rounds while others can. Again, a better buying price also comes with a higher chance of running away from a deal. We see that `GPT-4` achieves the best trade-off here: it gets the best price over multiple rounds with a higher success rate of reaching a deal.

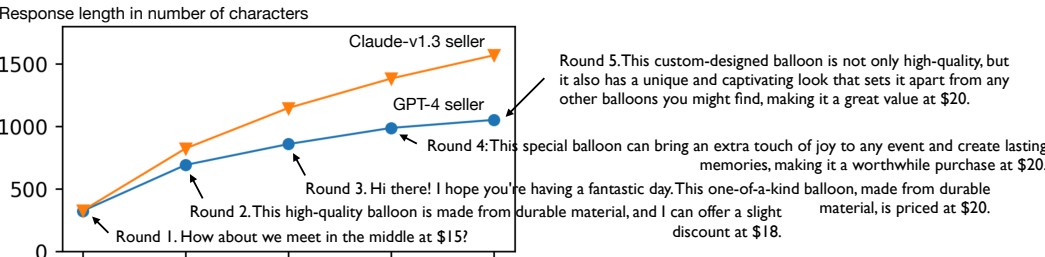

Figure 7: The average response length increases as the model learns from multiple rounds. Here we show examples of the seller's response when being asked the buyer's initial query "Would you consider selling it for $10?" After multiple rounds of negotiation, the seller's responses become more verbose and word-tuned. Yet verbosity does not mean better strategy: `claude-v1.3` is more verbose (higher curve) than `gpt-4`, but it has a worse success rate and deal price (recall Fig. 6). This indicates that `gpt-4`'s verbosity is more strategic.

of deal breaking. This suggests future research may consider global optimization for improving the overall gain over multiple rounds. We further show evidence of improved language from iterative AI feedback: in a multi-round game, one model may be less verbose than another, but be better word-tuned, thus more effective in getting a better deal.

We believe our results have profound implications for AI research: on the positive side, it indicates the possibility of continuously improving language models with minimal human intervention. On the risky side, it might be more challenging to oversight the model behavior in our framework because models are acting autonomously, which calls for future alignment and safety research in the multi-agent game setting. Overall, we believe our work provides a meaningful initial exploration for large language models' learning from game-playing and iterative AI feedback.

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
