# OpenReview forum: "Improving Language Model Negotiation with Self-Play and In-Context Learning from AI Feedback"
_NeurIPS.cc/2023/Conference — Submitted to NeurIPS 2023_

### Official Review · Reviewer_Tqe8 · 2023-07-05

**Soundness:** 3 good
**Presentation:** 3 good
**Contribution:** 2 fair
**Rating:** 5
**Confidence:** 3

**Summary:**

The paper studies the ability of LLMs to improve in a negotiation game. The find that only a subset of language models can self improve from AI feedback, a model's ability to learn from feedback depends on its role in the game, and stronger agents can go through more rounds of negotiation.

**Strengths:**

- The paper is generally well written; it was more or less easy to understand the entire paper.
- The experiment methodology is sensible. It was nice to narrow down the set of models by eliminating models based on their ability to respond to feedback or understand the problem.
- The results are interesting and could be valuable to the community on understanding the role of LLMs as agents.

**Weaknesses:**

- It seems like incorporating feedback is done by providing the feedback as context for the LLM. The paper could be made stronger by utilizing the fine-tuning APIs for the given models.
- Using GPT3.5 as the fixed agent is interesting. It's clear GPT-4 is the strongest agent in this scenario and it'd be interesting to see how well these results would hold against a stronger agent. In a similar vein, seeing how well these results would hold when actually negotiating with a human.
- Could buyer's and seller's be prompted better? E.g. would it be possible to prompt claude instant v1.0 more effectively to respond to multi-turn negotiations or prompt Cohere command or AI21 j2 better to respond to bargaining and feedback respectively?

**Questions:**

Overall, I think the work is interesting, it's reasonably well designed, and has interseting implications in reinforcement learning and large language models. I think there were some decisions (e.g. incorporating feedback in the context passed to the LLM, using GPT3.5 as the fixed agent that we are buying/selling against) that make the paper's results a little less clear. However, the results are interesting and show a first step towards using LLMs as agents, in negotiations.

**Limitations:**

The authors partially address limitations of their work, and address societal implications of their work.

---

> ### Author Rebuttal · Authors · 2023-08-10
>
> We thank the reviewer for the comments. Below we address the following suggestions:
>
> - “Utilizing the fine-tuning APIs for the given models”: we would very much love to do so, yet we do not have access to finetune GPT-3.5 / GPT-4 / Claude. Note that currently the general research and open-source community do not have access to finetune these models, and the current OpenAI finetune API is on GPT-3 (which is significantly weaker and does not understand the basics of the game), not GPT-3.5 / 4.
> - “Seeing how well these results would hold when actually negotiating with a human”: we would very much love to do so, yet to have statistically significant distribution of deal prices we need the negotiation to be replayed 500 times — in our experiments we did more than 10k negotiation runs. This scale is too large and expensive for us to do human experiments (even just 500 runs). Yet we also view this as a demonstration of the scalability of AI feedback (then human feedback), which is an advantage of our approach.
> - “Could buyer's and seller's be prompted better?”: when we started the initial experiments, we found that Claude and GPT can already follow the instructions well enough, so we did not further optimize the prompts for them. For cohere and AI21, since their model does not quite understand the rule of the game, we spent non-trivial level of efforts on prompt engineering cohere and AI21, yet not matter how we try we cannot make them understand, so we just report their failure behavior in our Figure 3.
> - “Incorporating feedback in the context passed to the LLM”: There are many existing / concurrent works showing that the feedback, in the format of natural language is effective and promising, see Scheurer et. al. 2022 and Madaan. et. al. 2023. Intuitionally, as long as the model has a basic level of instruction following, it can understand and follow natural language feedback. Our results align with existing works about language feedback, and we believe that AI feedback in the format of natural language is indeed a promising direction to further explore.
> - “Using GPT3.5 as the fixed agent that we are buying/selling against”: We made this decision given the limited budget constraint. Currently, GPT-3.5 is more or less the default baseline when comparing models, as is the practice in Zheng et. al. 2023 and Want et. al. 2023. So we follow this major practice. We would love to explore the comparison between more model should the budget permit.
>
> **References**
>
> Bai et. al. 2022. Constitutional AI: Harmlessness from AI Feedback
>
> Scheurer et. al. 2022. Training Language Models with Language Feedback
>
> Madaan. et. al. 2023. Self-Refine: Iterative Refinement with Self-Feedback
>
> Zheng et. al. 2023. Judging LLM-as-a-judge with MT-Bench and Chatbot Arena
>
> Want et. al. 2023. How Far Can Camels Go? Exploring the State of Instruction Tuning on Open Resources

---

### Official Review · Reviewer_sQTf · 2023-07-05

**Soundness:** 2 fair
**Presentation:** 3 good
**Contribution:** 1 poor
**Rating:** 5
**Confidence:** 4

**Summary:**

This paper investigates the intriguing possibility of autonomous improvement among multiple large language models through a negotiation game. By assigning various LLMs to distinct roles and allowing them to engage in iterative improvement, the paper aims to enhance their negotiation strategies without human intervention. The study uncovers some insights into negotiation problems, encompassing the assessment of model capabilities and their responses to AI feedback. However, after careful evaluation, I am inclined to reject this paper due to its limited contribution and insufficient experimental results, which fail to adequately demonstrate the effectiveness of its findings.

**Strengths:**

This paper studies an interesting problem: Improving large language models with each other and demanding only black-box access is a promising direction.
This paper is largely clear and concise. It is easy to follow the problem setting and the negotiation process.
This paper compares capabilities (especially the continue learning ability using in-context learning) of some advanced large language models in the proposed negotiation problem, which is interesting.

**Weaknesses:**

One of the main technical novelties is the AI feedback technique used in their method. However, this technique seems like a result of random attempts plus some intuition. More detailed though on how did the authors develop this technique or the comparison between other possible candidate techniques is needed. Moreover, this technique is similar to CoT. It would be better for the authors to discuss on the relationship between these two techniques. I am curious about the performance of guiding LLMs to think step by step without relying on additional critics in negotiation problems. More thorough explanations or experimental results of this aspect would provide deeper insights into the effectiveness of the proposed approach.
This paper aims to improve the ability of large language model by self-play with each other. However, to strengthen the paper's claims, it would be better to provide additional evidence regarding the transferability of the proposed framework to different types of games. Suggestions on the specific implementation process would also be valuable in applying this framework to different domains.
The stability of the environment is also a concern since the results appear to heavily rely on the reliability of the moderator. This dependence raises doubts regarding the individual contributions of various components in the system. Furthermore, the claim of proposing a technique for prompt optimization for generic classification tasks in Line 120 lacks sufficient details and evidence. Elaborating on this technique and demonstrating its effectiveness would enhance the paper's credibility and address this specific weakness.
It is not clear whether there exists an upper limit of improvement using ICL. It would be better for the authors to also discuss ICL with, e.g., fine-tuning or other trainable parts.

**Questions:**

What is the exact definition of "self-play"? It seems that only one of the two players improves in the setting of this paper, which slightly different previous definition in, e.g., https://arxiv.org/pdf/2002.04017.pdf.
Line 59: It would be beneficial to provide additional details, possibly with references, comparing the effectiveness of using AI feedback with RHLF. This comparison would contribute to a more comprehensive understanding of the proposed approach's advantages and distinguish it from existing techniques.
Line 85: Why the LM engine behind the critic is consistently the same as the player it provides feedback.
Figure 4: It lacks the experimental results of Claude-v1.3 and GPT-4 as buyers. Why these scenarios are omitted? Including these results in the figure would ensure a complete representation of the experiments conducted.



Presentation:

Line 118, 119: inspect -> inspecting, add -> adding
Line 120: recommend -> and recommending
Line 211: comparison -> comparisons
Figure 5: whild -> while
Ling 257: use less -> uses fewer
Line 258: serve -> serves

**Limitations:**

The authors discuss the limitations of the work in the last section.

---

> ### Author Rebuttal · Authors · 2023-08-10
>
> We thank the reviewer for the comments. Below are our responses:
>
> ## Significance of AI feedback and our contribution
>
> Our systematic investigation aims to answer the research question of whether multiple Large Language Models (LLMs) can improve each other in a negotiation game with minimal human intervention, and the negotiation game framework and the methods used in the game are designed to achieve this purpose.
>
> At the first glance, the settings of this work may look superficially easy (for real human), but are indeed nontrivial for models. This is why much of our effort is spent on designing the basic configurations of our game playing. Our method systematically converge to better results with quantitative evidence and qualitative examples, not just random attempts (as is demonstrated in our figure 4,5 and 6). The core contribution of this work is showing that LLMs, when strong and aligned enough, would be possible to continuously improve through self-play and AI feedback, as is summarized in our Figure 1C and detailed in our Figure 6. Qualitative examples about how the model improve over multiple rounds of game playing and AI feedback is shown in our Figure 7.
>
> ## Comparisons between in-context learning from AI feedback to other related approaches
>
> **In-context learning v.s. finetuning v.s. RL**
>
> - The reviewer asks us to “discuss ICL with, e.g., fine-tuning or other trainable parts” also “comparing the effectiveness of using AI feedback with RHLF”. Also we very much love to experiment SFT and RL, we do not have any finetuning / RL access to GPT and Claude.
> - In general, the current conclusion (from the collective experiences of the community) of the three learning paradigm is that ICL is cheaper than supervised finetuning (SFT) and cheaper than RL, which is why the current major paradigm of interacting with LLMs is ICL.
> - Currently, only very few big companies have the resource to do large scale finetuning and RLHF, and their results are that all three paradigms improve the model. The current major practice is always start from ICL to study what prompt data is effective, then apply SFT then RL (see Touvron et. al. 2023). Although ICL may not have the same improvements as SFT and RL, the improvements from ICL can still be nontrivial given the correct prompt (as is shown in our Figure 7) and the data used in ICL is usually also effective when used for SFT (e.g., Fu. et. al. 2023).
>
> **AI feedback v.s. chain-of-thought**
>
> - We find it very hard to understand why the review believe AI feedback is similar to chain-of-thought. To the best of knowledge, also we believe that the majority opinion of the research community is that these two are parallel ideas and most existing work only focus on one of them, like AI-feedback-only in Bai et. al. 2022 and Scheurer et. al. 2022, or chain-of-thought-only in Wei et. al. 2022. There is very recent work that tries to combine the two technique together like Madaan. et. al. 2023, but they treat the two technique complementary, rather than one as the alternative to the other.
> - In our case, we could also combine the two technique together, like having the player to think about what to say before actually saying it, then have the critic to provide feedback to the player’s internal thinking process. We believe this is an interesting direction to explore and will try it in the next followup work.
>
> ## Other important concerns
>
> **Implementation and transferability of our experiments**
>
> - We will open source all our prompts and codes to support research in the direction of game playing and AI feedback.
> - Further about the applicability and effectiveness of AI feedback on other domains, we note that there are concurrent work showing in-context learning from AI feedback is also effective for reasoning (Madaan. et. al. 2023) and factuality (Du el. al. 2023). Combining our results with these two concurrent works, we believe that AI feedback is indeed an effective and promising direction for improving LLMs.
>
> **Claude v1.3 and GPT-4 as buyers**
>
> - The performance of Claude v1.3 and GPT-4 as buyers are shown Figure 6, B1, where the GPT-4 buyer can consistently improve over multiple rounds, and the Claude-v1.3 buyer improves until the third round. The price distribution of these two engines are similar to the distribution in Figure 4 and 5, so we omit them to save space.
>
> **Why using the same engine for the critic and the player**
>
> - Since the overall study of game playing and AI feedback in the research community is still in a relatively early stage, an important objective of this work is to set up the basic experimental settings before pushing it more complex, which requires us to focus on the major factor we want to study (which is AI feedback) while keeping other factors minimal and constant. This is why we use the same engine for the critic and the player to keep the setting simple. We will try to more settings if budget permitts?
>
> **References**
>
> Fu et. al. 2023. Specializing Smaller Language Models towards Multi-Step Reasoning
>
> Yuan et. al. 2023. Scaling Relationship on Learning Mathematical Reasoning with Large Language Models
>
> Touvron et. al. 2023. Llama 2: Open Foundation and Fine-Tuned Chat Models
>
> Bai et. al. 2022. Constitutional AI: Harmlessness from AI Feedback
>
> Scheurer et. al. 2022. Training Language Models with Language Feedback
>
> Wei et. al. 2022. Chain-of-Thought Prompting Elicits Reasoning in Large Language Models
>
> Madaan. et. al. 2023. Self-Refine: Iterative Refinement with Self-Feedback
>
> Du el. al. 2023. Improving Factuality and Reasoning in Language Models through Multiagent Debate
>
> Gao et. al. 2022. Scaling Laws for Reward Model Overoptimization

---

> > ### Comment · Reviewer_sQTf · 2023-08-21
> > **Response to the rebuttal**
> >
> > Thanks to the authors for the detailed response. I appreciate the additional clarifications from the authors.

---

### Official Review · Reviewer_nXyn · 2023-07-06

**Soundness:** 3 good
**Presentation:** 3 good
**Contribution:** 3 good
**Rating:** 4
**Confidence:** 4

**Summary:**

This paper studies whether multiple large language models (LLMs) can improve each other in a negotiation game by playing, reflecting, and criticizing, with minimal human intervention. Two LLMs play the roles of a seller and a buyer, and a third LLM plays the role of a critic who provides feedback to one of the players to improve their negotiation strategy. The authors report several intriguing findings, such as the different abilities and behaviors of various LLMs in the game, the trade-off between deal price and success rate, and the evidence of improved language complexity and strategy from iterative AI feedback.

**Strengths:**

1. Interaction between LLMs is quite interesting and potentially important for future AI research.
2. The authors experimented with a variety of LLMs, including open-source (Cohere) and proprietary (GPT, Claude) ones.

**Weaknesses:**

Major Issue:
1. The negotiation setting is quite contrived and not well-grounded. Specifically, there is no context about the goods being discussed. Also, the feedback and overall conversation seems very generic (Figure 2). There is no value/intrinsic motivation for the buyer to get the goods, and there is only one choice to go for. Typically, negotiation in real-life doesn't happen this way.
2. It feels like the current LLM negotiation setting can be viewed as `predicting the most probable outcome/text' given the negotiation contexts.
3. AI feedback is not the same as human feedback, the behavior may be completely different in real-world situations since there are a lot of other concerns including 'value', 'time', 'personal preference' for negotiation. This makes me wonder what this simple negotiation has to offer in terms of understanding/future research.


Minor Issue:
1. The title is a bit misleading. It seems to me from the title that the paper is about an algorithm, but it is more about evaluations and understanding.


[After Rebuttal]

My concern still lies in the 'motivation' behind the setting for the experiment. In my understanding, negotiation requires a setting, e.g., both seller and buyer will have a stake on the product (buyer want to pay lower than the *utility* of the product and seller want to earn more than the *cost*). However, such basic setting is not present in the experiments.

This makes me concerned if the observed effect is really negotiation or just the effect of instruct fine-tuning.

I decided not to change my score.

**Questions:**

1. I wonder why LLAMA based models are not considered in this work.
2. Is it possible to have both the seller and buyer receive feedback in a single experiment? I wonder what will happen.
3. Is it possible to add some story/context to the negotiation so that the LLMs have a better motivation?
4. Is the best policy to just always be fixed at starting price and ask LLM to generate some excuse (Prompt: "Give me some sentences to sell my __ for the highest possible price.")?

[After Rebuttal]

All questions have been addressed.

**Limitations:**

I was unable to find any information related to limitations.

---

> ### Author Rebuttal · Authors · 2023-08-10
>
> We thank the reviewer for the comments. The reviewer is mostly concerned the design of our game setting and asks if our setting can be extended add more factors like “context” “motivation”, “personal perference” and so on.
>
> We would like to note that, although we very much love to study how these factors influence the behavior of language models, we need to set up the basics before pursuing more complex settings. The focus of this work is to study whether it is possible that LLMs can improve each other through AI feedback **while keeping other orthogonal factors minimal and constant**. Specifically:
>
> ## We need to set up the basics before adding more factors
>
> The reviewer suggested to add more:
>
> - “context about the goods”
> - “value / intrinsic motivation”
> - “value”, “time”, “personal preference”
> - “story / context”
>
> These are all great direction that we very much love to explore. However, practically, there are two important constraints that require us to set up the basics first, specifically:
>
> - At the current stage, some model engines cannot even understand the basic commonsense like \\$8 is a lower price than \\$10 (Figure 3). **How can we ask them to incorporate complex ideas like “value” “personal preference”, “motivation” when they don’t even understand the very basics like \\$8 is lower than \\$10**?
> - In our experimental setup, we carefully disentangle / rule out factors that is orthogonal to the primary factor (AI feedback) we study. This work’s goal is to study how AI feedback can improve the deal price while **controlling other confounding factors to be minimal and constant**. This is why we intentionally set other factors to be simple and minimal, such that we can focus on how different LLMs incorporate AI feedbacks.
>
> Overall, we would love to test all the factors suggested by the reviewer in the next follow up work, yet for this work, we need to carefully set up the minimal basics (which is already quite nontrivial) before making it more complicated (which is meaningful, but orthogonal to our focus).
>
> ## Other important concerns
>
> “why LLAMA based models are not considered in this work”
>
> Because when we started this project, there was no good LLaMA based model that could understand the basic rule of bargaining and make similar mistakes shown in our Figure 3. This is also why we choose the four model families discussed in Figure 1C.
>
> “Is it possible to have both the seller and buyer receive feedback in a single experiment?”
>
> Yes, we did that at the initial runs, but we observed the deal price will be fixed at $15 because the two players improves simultaneously, making it hard to evaluate the effectiveness of AI feedback.  Setting one player fixed while the other receiving feedback enable us to evaluate the progress using the deal price.
>
> “Is the best policy to just always be fixed at starting price and ask LLM to generate some excuse (Prompt: "Give me some sentences to sell my __ for the highest possible price.")?”
>
> We are not sure what the reviewer means here. In our practice:
>
> - We always set the initial starting price for the seller to be \\$20 and for the buyer to be \\$10, such that we can control the price range to be within [10, 20], thus being able to measure the progress using the final deal price. If this is what the review mean by “always be fixed at starting price” then we have already done it.
> - Our initial prompt to the seller involves “The cost of your balloon is \\$COST_PRICE and your starting price is \\$SELLER_INIT_PRICE. Your goal is to sell it to a high price.” if this is what the reviewer mean by “Prompt: Give me some sentences to sell my __ for the highest possible price.” then we have already done it at the very beginning. Also note that the AI feedback, when applied to strong and well aligned agents like GPT-4, will continuously improve the deal price over this initial prompt, suggesting simply asking the model to “sell it to a high price” is not the optimal strategy.
> - Note that for the seller, never lowering the price is not the best strategy because it often fails to reach any deal -- as is shown in our Figure 6, selling too hard comes with a higher risk of breaking the deal.

---

> > ### Comment · Reviewer_nXyn · 2023-08-13
> >
> > Thank you for the detailed responses. Overall, I think the setting of the experiments is not solid enough:
> >
> > 1. Use of GPT-3.5/GPT-4 as evaluator. If GPT-3 makes a mistake then the results may not be accurate.
> > 2. Meaningless negotiation. There is no context for the LM seller for the 'cost' of the good and no context for the buyer for the 'utility' in the economic model. There is no practical reason where the price should be set. This makes the task more like some NLP benchmark.
> > 3. Limited LLM formats. Different LLMs may behave differently under different prompts and different LLMs may also behave differently. Therefore, I'm concerned how instructive this work would be for future works.
> >
> > Overall, I would retain my score for the paper. Please let me know if there are any further misunderstandings.

---

> > > ### Author Response · Authors · 2023-08-13
> > >
> > > We thank the reviewer for the response. We would like to further clarify certain misunderstandings:
> > >
> > > - **We do not use GPT-3 as evaluator**: there is no “GPT-3” in this paper — all experiments we use GPT-**3.5** and GPT-**4**. We use the **final deal price as the evaluation** (see our Figures 2, 4, 5, 6), which is a standard practice of AI bargaining research (e.g., see Heddaya et. al. 2023). We are well aware of how GPT-based evaluation can be biased (e.g., see Wang et. al. 2023), this is why we choose the evaluation to be the final deal price, as it is more objective.
> > > - **Our experiments clearly show how the negotiation improves in multiple meaningful ways**. The value of the product is discussed multiple times during negotiation, and we give many examples in the paper (e.g., Figure 2C: “high quality latex and handcrafted by expert artisan”; Figure 7 “made from durable material”)
> > > - **The prompts, especially after AI feedback, is of very high diversity.** We note that one should not view the initial instruction as the fixed prompt (we keep the initial instruction fixed because we need to use the same way to explain the rule of bargaining to different agents), but should **view the [initial instruction + round 1 dialog + round 1 AI feedback + round 2 dialog + round 2 AI feedback … ] collectively** **as the prompt** for the next round negotiation. To give a sense of how diverse the prompts are, in Figure 6, we run the game 5 rounds, and repeat 500 times, which in total gives 2500 different types of prompts. Because our prompts are so diverse, we observe multiple meaningful behaviors and a wide range of deal price distributions. We show some of the examples in Figure 1B where we sample four types of AI feedback, as the updated prompt for the player, and how it improves players’ strategy.
> > >
> > > References
> > >
> > > Heddaya et. al. 2023. Language of Bargaining
> > >
> > > Wang et. al. 2023. How Far Can Camels Go? Exploring the State of Instruction Tuning on Open Resources

---

> > > > ### Comment · Reviewer_nXyn · 2023-08-14
> > > >
> > > > Thank you for the clarifications. I have adjusted my comment to better reflect accuracy.
> > > >
> > > > By GPT-3, I meant language models in general

---

> > > > > ### Author Response · Authors · 2023-08-14
> > > > > **we would appreciate reviewer nXyn could keep their original comments instead of deleting / modifying them after we have addressed them**
> > > > >
> > > > > We thank the reviewer nXyn for their response. About evaluation, we would like to re-iterate:
> > > > >
> > > > > - **We do not use any GPT/ LLM as the evaluator**. We are well-aware the drawbacks of using LLM as evaluators, as is discussed in concurrent works like Wang et. al. 2023.
> > > > > - **We use the final deal price as the evaluation**, as the common practice of bargaining literature, because it is more objective.
> > > > >
> > > > > Additionally, **we would appreciate reviewer nXyn to keep their original comments instead of deleting / modifying them after we have addressed them**, specifically:
> > > > >
> > > > > About whether the players are having meaningful negotiation, initially the reviewers wrote
> > > > >
> > > > > > “There is no context for the LM seller for the 'cost' of the good and no context for the buyer for the '**value**' in the economic model”
> > > > > >
> > > > >
> > > > > **After we have addressed how the value of the product is discussed in the above response, the reviewer deleted the word 'value'** and changed their comments to be
> > > > >
> > > > > > There is no context for the LM seller for the 'cost' of the good and no context for the buyer for the '**utility**' in the economic model
> > > > > >
> > > > >
> > > > > Similarly, for the diversity point, initially the reviewer writes:
> > > > >
> > > > > > “Limited LLMs/prompt formats”
> > > > > >
> > > > >
> > > > > **After we have clarified the diversity of the prompt, the reviewer deleted the word 'prompts'** and changed this to be
> > > > >
> > > > > > “Limited LLM formats”
> > > > > >
> > > > >
> > > > >
> > > > > The reviewer’s original post can be found in the OpenReview’s notification email.
> > > > >
> > > > > Further on the reviewer’s updated comments:
> > > > >
> > > > > - We would like to note again that at the current stage, some model engines cannot even understand the basic commonsense like \\$8 is a lower price than \\$10. **How can we ask them to incorporate “the 'value' in the economic model” when they don’t even understand the very basics like \\$8 is lower than \\$10**?
> > > > > - About the concern about “limited LLM formats”, **we have already considered most leading LLMs** (Cohere, AI21, Claude-Instant-1.0, Claude-1.3, GPT-3.5, GPT-4) at the time of our experiments, note that although there are many open-source LLaMA based models, they do not exist when we started our experiments

---

> > > > > > ### Comment · Reviewer_nXyn · 2023-08-14
> > > > > >
> > > > > > Hello,
> > > > > >
> > > > > > Thank you for organizing the history/edits.
> > > > > >
> > > > > > Please note that I have always meant ‘value to the buyer’ or utility starting with the first version of my review.

---

> > > > > > > ### Author Response · Authors · 2023-08-14
> > > > > > >
> > > > > > > We thank the reviewer for the response.
> > > > > > >
> > > > > > > We understand that the reviewer has been insisting on adding “value to the buyer” or ”utility” as well as other factors like “context about the goods”, “intrinsic motivation”, “personal preference” .etc. These are great directions that we very much love to explore. However, as we have been emphasizing, **how can we ask the model to incorporate such complex ideas when they don’t even understand the very basics like \\$8 is lower than \\$10?**
> > > > > > >
> > > > > > > The development of LLMs is simply not that advanced yet. We would like to note that at the current stage of science, it is more important to set up the basics to test whether LLMs can autonomously improve each other through game playing and AI feedback in a minimal setting, which is precisely the contribution of this work.

---

> > > > > > > ### Comment · Reviewer_nXyn · 2023-08-14
> > > > > > >
> > > > > > > Hello,
> > > > > > >
> > > > > > > I understand that the deal price is the evaluation metric, but doesn’t the determination of the deal price involve an LLM?
> > > > > > >
> > > > > > > Also, in regards to value/utility. I regard the utility or value to the buyer as a key to making a deal, similar to the cost to the seller. If the argument is that we cannot include any context before we start negotiating because some LLMs do not understand context, then then it is unclear how much of the whole negotiation process the LLM understands

---

> > > > > > > > ### Author Response · Authors · 2023-08-14
> > > > > > > >
> > > > > > > > We thank the reviewer for the continued discussion. Below is further clarification:
> > > > > > > >
> > > > > > > > "but doesn’t the determination of the deal price involve an LLM?"
> > > > > > > >
> > > > > > > > * We use the language model to classify whether the two players have reached a deal. This classification has achieved 90+ accuracy after extensive prompt engineering.
> > > > > > > > * Then we extract the actual price number by regular expression -- simply extracting the last number in the sentence (also the practice in many existing works using numbers as metrics, e.g., in Fu et. al. 2023) and it is accurate enough -- we generally believe this part is simple and can be achieved by any researchers who have basic experiences of prompt engineering.
> > > > > > > >
> > > > > > > > Regarding the motivation for making the deal
> > > > > > > > * **We instruct the model to buy / sell the product to a lower / higher price** by putting "Your goal is to buy it with a low price, as low as possible" in the initial instruction.
> > > > > > > > * Because the models are trained to follow human instructions, we tend to believe our **human instruction is their largest motivation / the key for them to make a deal, not their own interest** -- or in other words, they are more or the less doing the bargaining on because they are trained to follow human instructions that ask them to do so.
> > > > > > > > * Again, we very much love to test how the model will behave if they bargain for their own interest, but that is beyond this work's scope -- our objective is to set up a minimal testbed to study if language models, under human instructions, can self-improve over AI feedback
> > > > > > > >
> > > > > > > > We hope this clarifies the misunderstandings.
> > > > > > > >
> > > > > > > > References
> > > > > > > >
> > > > > > > > Fu et. al. 2023. Complexity-Based Prompting for Multi-Step Reasoning

---

### Official Review · Reviewer_SEMS · 2023-08-01

**Soundness:** 2 fair
**Presentation:** 2 fair
**Contribution:** 2 fair
**Rating:** 4
**Confidence:** 4

**Summary:**

This paper studies the strategic multi-agent problem setting of two LLM Players interacting in a negotiation (or bargaining) game and proposes to use feedback from an LLM Critic to improve each Player’s expected behavior and performance in the game.  Importantly, the paper aims to study how AI Feedback can enable Player improvement when playing competitive games under well-defined rules.  The proposed method is to use the player dialog history and critic feedback as “in-context demonstrations” for players to self-improve over the course of the game.  Experiments are conducted on a bargaining game instance negotiating the price of a balloon, investigating several LLMs as base models for the Players (and their Critics).

**Strengths:**

Identified strengths of paper:
 - I really like the premise of this work: that agents playing in a strategic game may each be able to benefit and improve their negotiation strategies, not only by observing their opponents, but additionally by heeding the advice of personalized critics whose aim is to help (cooperate with) them.  Like other AI feedback methods, this type of approach has the potential to scale better than using Human feedback for the same type of assistance, presuming feedback is helpful and thus desirable for this negotiation task.
 - The topic is timely and very relevant to the multi-agent and game theoretic communities, as it takes a long-existing and well-studied problem (multi-agent negotiation) and investigates how rational player strategies can be improved using AI feedback and LLMs.


**Weaknesses:**

Identified concerns and suggestions to improve the paper submission:
- It would be extremely helpful to see examples of what the *instructions* and *in-context learning* looks like for the Critic agents? Examples of their few-shot prompts (or instruction data for Finetuning) are important, as this is a key contribution of the work (adding AI Feedback to strategic reasoning/negotiation settings). Notably, where are the Buyer/Seller Critics getting the suggested negotiation strategies in Figure 1B  from (e.g. the flinch technique, the anchoring technique, etc)?  They seem to be good, general strategies to have in the player’s toolbox, but require the Critic to have knowledge of effective negotiations and how/when to employ them.  Thus, how are the Critics coming up with general negotiation strategies and reasoning about when to apply them?  This is important for opening the black box of the AI Feedback component of the proposed method.
 - While I like the overall premise, the novel technical contribution of this work seems tenuous.  With that, the problem setting is strongly inspired by AlphaGo Zero, but is there any traditional learning in this setting (i.e. any updates to the agent policies based upon the AI feedback given)?  That is still unclear to me.  Currently the paper seems to simply add a small amount of additional context in the prompt for the Player LLMs.  If there is no traditional learning, this setting is also critically different from AlphaGo Zero in that the proposed feedback signal (from Critic Agents) is **not** used to update the Player policies.  This should be explicitly clarified.  Also, if there are model updates, it would be useful to see the learning update rules.  If not, why not do any finetuning of the model?  At least as an experimental condition to compare against and understand if in-context learning is "sufficient".  Perhaps more motivation and context for this design decision would be helpful.
 - This work explores a potentially interesting research direction (negotiation games or strategic gameplay more generally + AI Feedback for improved player strategies) but it’s not clear to me how well motivated AI feedback is for this negotiation game or how interesting the problem is in its current instantiation.  In particular, it would be helpful if the paper provided some type of performance analysis to show that the negotiation game used (an instance of balloon purchasing) is interesting and challenging to solve *before* ever adding in LLMs or a Critic.  How would existing multi-agent negotiation/bargaining approaches solve this game?  What equilibria do the Players generally converge to without LLMs?  How and why does using LLMs change the *expected* solution?  Given two Player LLMs, why is a Critic *necessary* to have?  In other words, if the Critic is being prompted with examples of bargaining conversations, could the Buyer/Seller agents not simply see those same examples and then converge upon the same solution (they are currently finding) *without* a Critic?  Or is there some value that the Critic role *uniquely* adds (e.g. more efficient or robust convergence on an equilibrium in this negotiation game)?  Motivating the game selected as an interesting and challenging problem in its own right is critical.
 - Furthermore, I still question the generality and significance of the paper’s empirical findings.  **RE Significance:** How significant and meaningful are the differences in Seller performance in Table 1?  The numbers don’t seem that different to me, but perhaps with more context, the significance becomes more clear.  With that, I see a distributional shift in Figure 4, but how meaningful is this shift?  What are the implications of it, regarding how good or bad these solutions/equilibria are before and after receiving AI feedback?  Do the solutions to the negotiation game simply change a small amount but are comparable in terms of how preferable/desirable they are or do they get qualitatively better in some way?  **RE Generality:** The paper seems to use only one evaluation domain (balloon purchasing).  It is not clear to me which empirical findings/trends from the use of AI Feedback on this one, seemingly simple bargaining domain, are expected to generalize. In particular, regarding the effectiveness and impact of AI Feedback in other multi-agent negotiation domains and settings.  This limited evaluation seems more like an interesting case study than general findings that transfer.
 - In Subsection 4.3, the paper provides analysis of each of the LLM models (gpt-3.5, gpt-4, claude-v1.0, etc) versus a fixed gpt-3.5-turbo opponent.  However, a more thorough investigation examining the cross product of each LLM model as Buyer against every other LLM model as Seller (a *complete* Payoff matrix with all pairwise model comparison) could potentially provide more general insights. Why was this not done? Can it still be done to show a resulting Payoff Matrix?  Not doing a pairwise comparison of *all* pairs of models might unnecessarily constrain the results and thus the insights that can be extracted.

**Questions:**

Please see Weaknesses section.

**Limitations:**

Please see Weaknesses section.

---

> ### Author Rebuttal · Authors · 2023-08-10
>
> Overall, the design decisions that we make are closely based on our understanding of what the current LLMs can do, and how to push it further. We aim to clarify the following points:
>
> - **Policy updates by In-context learning from AI feedback**: the reviewer discussed that our approach “simply add small amount of additional context” and “the feedback signal is not used to update the policy”. We tend to believe this is an important misunderstanding and beg to differ: it is precisely the small amount of AI feedback, used as the new instructions to the model, that changes the model’s policy drastically (given the model is strong and aligned enough), as is show quantitatively in our Figures 4,5,6, and qualitatively in Figure 7. Similar observations are made in Madaan et. al. 2023 and Shinn et. al. 2023.
> - **Why not doing fine-tuning and what about without LLM**: unfortunately, our setting is actually very challenging for models weaker than Claude-instant, as is shown in our Figure 3. We have also tried some open-source models, or methods before LLM, yet they struggle to understand the very basic common sense like \\$17 is a higher price than \\$15. One important message from this work is the model has to pass a certain bar to even play the game. At the current stage of research, we tend to believe only the GPT and Claude family can play our game and improve over rounds, which we do not have access to finetune.
> - **How the critic can come up with suggestions**:  In general, if the model engine is better than or equal to Claude-instant, such as GPT-3.5 and Claude-1.3 (Figure 1C), it has no difficulty writing meaningful critics (which is also the observation of Madaan et. al. 2023 and Bai et. al. 2022). How inside the neural network the model figures out a feedback is still a challenging research problem, and our observation is that the model usually write critics according to the context. For example, if it observe the player commits to a price too early in the previous round, it tends to ask it to stand firm in the next round.
>
> We further clarify how AI feedback updates the player’s policy, specifically:
>
> - **Evidence about player’s policy updates from AI feedback**: when the model has a certain level of ability to understand the follow instructions, it will adjust its strategy based on the critic’s suggestion. One direct evidence is figure 7, where after rounds of AI feedback, the player becomes more eloquent, word-tuned, strategic with the starting price, and emphasizing the product quality.
> - **The AI feedback, as the prompt to the model, is the key to trigger updated policy**: In this setting, the prompt is extremely important and far more than just “small additional context”. Figure 2 is an example/ evidence where the model respond to the suggestions by the critic where the critic gives three suggestions and the player follows two of them.
> - **In-context learning (ICL) is nontraditional but effective and widely deployed**: our setting can be viewed as using previous round dialog history + AI feedback as in-context learning demonstrations (given the model has the ability to follow AI feedback). ICL is indeed different than traditional gradient-based finetuning, but nowadays it is a widely deployed learning paradigm that can be effectively used for modifying model policy/ behavior. The mechanism of ICL is still an open research problem, and there are more evidence showing its ICL is equivalent to implicit gradient descent (Oswald et. al. 2023, Dai et. al. 2023) — from this perspective, our approach can be understood as using the in-context demonstration data to “finetune” the model such that it has better bargaining strategy in the next round of negotiation.
>
> The reviewer also has concerns about the significance and generality about the experimental results. To respond this,
>
> - For each setting of the game we repeat it 500 times to ensure the deltas and distributional shifts are statistically significant and consistent.
> - We note that the model’s behavior change will have a distribution shift over one round of the game (Figure 4) where they immediately use the strategies suggested by the critic (Figure 1B and 2).
> - The bargaining policy change will become more prominent in a multi-round game setting (Figure 5, 6), and qualitatively will become more and more word-tuned and strategic, as is shown in Figure 7.
>
> Further we address the reviewers’ other important concerns
>
> - **Why only comparing to GPT-3.5 turbo:** Because GPT-3.5 is nowadays the default baseline for comparing language models, as adopted by many works like Zheng et. al. 2023 and Touvron 2023. Comparing all pairs of models are just too expensive for us to run (the GPT-4 experiments already cost a fortune). That being said, we agree that a payoff matrix is indeed meaningful to compare multiple models, which we will add in the updated version of our paper.
>
> **References**:
>
> Madaan et. al. 2023. Self-Refine: Iterative Refinement with Self-Feedback.
>
> Bai et. al. 2022. Constitutional AI: Harmlessness from AI Feedback.
>
> Shinn et. al. 2023. Reflexion: Language Agents with Verbal Reinforcement Learning
>
> Zheng et. al. 2023. Judging LLM-as-a-judge with MT-Bench and Chatbot Arena
>
> Touvron et. al. 2023. Llama 2: Open Foundation and Fine-Tuned Chat Models
>
> Oswald et. al. 2023. Transformers learn in-context by gradient descent
>
> Dai et. al. 2023. Why Can GPT Learn In-Context? Language Models Implicitly Perform Gradient Descent as Meta-Optimizers

---

### Decision · Program_Chairs · 2023-09-21

**Decision:**

Reject

**Comment:**

This paper takes a notable step in enhancing language model negotiation strategies through AI feedback, grounded in a solid experimental approach. However, it fails to fully elucidate the role and innovation brought by the critic agents, leaving critical questions on the utility of the AI feedback system in the negotiation setup. Moreover, the limited empirical scope raises concerns about the potential generality of the findings. Nevertheless, this is a very difficult decision, and the meta-reviewer believes that with further refinement and broader analysis, this paper could make a significant contribution in a future venue.